# Promoting Healthy Behaviors in Older Adults to Optimize Health-Promoting Lifestyle: An Intervention Study

**DOI:** 10.3390/ijerph20021628

**Published:** 2023-01-16

**Authors:** Fan Chia, Wei-Yang Huang, Hsuan Huang, Cheng-En Wu

**Affiliations:** 1Office of Physical Education and Sport, National Chung Hsin University, Taichung 402227, Taiwan; 2Physical Education Leader, National Taiwan College of Performing Arts, Taipei 11464, Taiwan; 3Department of Occupational Therapy, National Cheng Kung University, Tainan 701401, Taiwan; 4Office of Physical Education, Tamkang University, New Taipei City 251301, Taiwan

**Keywords:** healthy diet habits, regular physical activity, health responsibility, social support, smart bracelet

## Abstract

Introduction: Exercise intervention is the easiest and most effective way to promote human health. This study combined technology and exercise to improve the health behavior of the older adults through a physical activity intervention and to enhance a health-promoting lifestyle. Materials and methods: A quasi-experimental research method was used to openly recruit 120 healthy male and female older adults over 65 years old (average age of males: 71.6 ± 1.25 years; average age of females: 72.3 ± 1.28 years), all of whom wore smart bracelets. The participants were monitored by special personnel during the same period of walking every Monday to Friday. All participants recorded their daily steps, distance walked, and calorie consumption data for a period of 8 weeks. Results: After 8 weeks of walking, all participants showed a positive medium–high correlation of various factors between healthy behaviors and the health-promoting lifestyle scales. In the post-tests of each factor of two scales, males had the highest correlation between regular physical activity and physical activity, and females had the highest correlation between regular physical activity and social support. The variabilities in the explanatory power of the health behaviors of males and females on the health-promoting lifestyle were R^2^ = 70.9% (*p* < 0.01) and R^2^ = 74.1% (*p* < 0.01), indicating that the variables of healthy behaviors have a positive effect on health-promoting lifestyles in male and female older adults. Conclusions: Walking interventions positively affect the health behaviors of older adults and encourage health-promoting lifestyles. The value of this study is in its contribution to health promotion and public health recommendations for older adults.

## 1. Introduction

The World Health Organization (WHO) points out that unhealthy behaviors and the long-term effects of lifestyles can cause diseases. The cost of investing in health prevention is much lower than the cost of treatment after the disease occurs. Some older adults have begun to carry out auxiliary self-health prevention, management, and even rehabilitation through behavioral methods, consciously or unconsciously. In recent years, and in response to this need, the WHO has already announced that the concept of health promotion is the most effective way to promote human health [1]. According to the important key facts presented by the WHO, in 2020, the number of older adults over 60 years old will outnumber children under 5 years old, and the world population over 60 years old will increase to 22% by 2050 [2]. The United Nations (UN) General Assembly declared 2021–2030 the UN Decade of Healthy Ageing [3], during which promoting the healthy lifestyles of older adults will play an important role.

Population aging will have a comprehensive impact on health and medical care, the economy, education, social development, and welfare; obesity in older adult patients is associated with many diseases, such as cardiovascular disease, hypertension, and type 2 diabetes, and the morbidity and mortality rates also increase [4]. Experts believe that obesity is related to many factors, some of which may be caused by physical inactivity [5]. Physically active older adults have a reduced risk of all-cause and cardiovascular mortality, fractures, falls, cognitive decline, dementia, Alzheimer’s disease, and depression, and have better quality of life and cognitive function [6,7,8]. Chronic diseases are the main health problems affecting older adults, and improper control will lead to comorbidities, causing permanent harm to patients and increasing the burden on national healthcare systems [9,10]. The medical needs of older adults are far higher than those of adults of other age groups, so disease prevention and health promotion for older adults are becoming more and more important. The goal of disease prevention and health promotion for older adults is to minimize the damage caused by diseases and to maintain physical function and independent living [2].

Health promotion is a preventative approach to health that is self-actualization-oriented and guides individuals to maintain or improve their health and actively establish new positive behaviors [11,12]. Many older adults over the age of 85 require medical attention from doctors and paramedics, which is not actually a healthy behavior [13]. The healthy behavior of older adults should include exercise, smoking cessation, and the restriction of alcohol consumption, as well as increasing learning activities and social activities in life [14,15,16,17]. Some experts in the medical field firmly believe that the improvement in the health of older adults will not be achieved by taking drugs alone, but through a healthy lifestyle [18,19,20]. A so-called bad lifestyle consists of improper health behavior. Health behavior is defined as an individual engaging in behavior that positively or negatively affects their own health [21]. For example, a healthy diet and exercise are positive health behaviors [22], while negative health behaviors are smoking, excessive alcohol consumption, and risky sexual behaviors [23]. Scholars have pointed out that health promotion refers to the enhancement of a person’s physical, mental, and social well-being through any behavior, social conditions, and environments that can support their physical and mental health, strong mental state, longevity, and good quality of life. This is where a healthy lifestyle comes into play [24,25].

Reducing sedentary behavior and increasing physical activity in older adults can improve physical function, reduce stress, and improve quality of life [2]. The general public can understand that the aim of exercise interventions is to promote health, but many do not consider that people can use technology in combination with exercise. Individuals can detect their own exercise volume and intensity during exercise without the need for companions and you can also determine their own exercise volume and intensity during exercise. Only the exercise field can reach the limit of the exercise effect, which is the new typology of the combination of exercise and technology [26]. Valenti et al. pointed out that walking is the main physical activity of the older adults [27]. It was generally believed that walking was the most suitable physical activity for older adults, and the quantification of walking can be expressed in steps or distance traveled, in addition to walking time. The intensity of walking activities can be expressed by heart rate [28], and it was recommended that older adults achieve at least a moderate-intensity walking pace for 150 min [27,29]. There are more and more studies on artificial intelligence (AI) technology products, using smart bracelets for health and health care applications, monitoring body temperature, and in adult sports [30,31,32,33]. Lao et al. confirmed that wearing a sports bracelet can improve the motivation of physical activity in older adults [31]. Many studies have confirmed that the use of pedometers could accurately control the behavior of walking in real time and could promote walking tools [34,35,36], but there is not much research on improving older adults’ sports or physical activity monitoring. In accordance with the above literature, this study adopts a new typological point of view, combining the use of a smart bracelet with a physical activity intervention in older adults. Therefore, the purpose of studying a walking intervention in older adults, designed to change their health behaviors, was to optimize their health-promoting lifestyles. This research formulates the following research hypotheses. Hypothesis one: the intervention of walking helps older adults to promote healthy lifestyles. Hypothesis two: healthy behaviors after the walking intervention could effectively predict health-promoting lifestyles in older adults.

## 2. Materials and Methods

In this study, male and female older adults were divided into two groups: the pre-test group and the post-test group, each being measured twice. The trial was designed as a quasi-experimental study [37], and physical activity interventions were implemented between the pre-test and post-test.

### 2.1. Research Participants

This study took older adults from Taipei City as the study population (the content includes QR code link registration information, online registration, and printed registration brochures in community service centers), and 120 healthy older adults over 65 years old (60 males and 60 females) (according to the WHO’s definition of healthy older adults as those who can maintain normal life function ability and general cognitive level [38]) whose physical activity on weekdays was low (low physical activity is defined as never or almost never engaging in moderate or vigorous physical activity [39]). All studies included monitoring, detection, data collection, and report writing by four researchers from the beginning to the end. The whole event was free, and each participant could receive two free SPA rolls (due to free will, people who could not participate due to environmental, identity, or social and economic reasons were excluded). The participants were asked to intensively walk at a fixed time every Monday to Friday for 8 weeks (a total of two months, with on-site monitoring); Saturdays and Sundays were managed by the participants, but their steps and calories were still recorded. In terms of sample use precision, the mean difference of the sample size parameter (e.g., age, weight, height, body mass index (BMI)) was within the credible interval, and the sample size was consistent with the sample estimates for exercise science research [40]. The recruited participants were first checked for personal background information, including gender, age (years), weight (kg), height (cm), previous occupation, and current residence status (excluding those who lacked adequate decision-making ability due to age, intelligence, or physical condition or those who were vulnerable to undue influence and coercion or who were unable to participate due to their environment, identity, or social and economic conditions). All participants in this study signed an informed consent form, in line with scientific and ethical principles (contents include: no orthopedic disease or heart disease, not already participating in an exercise program, and willing to abstain from taking supplements that can increase muscle growth during the study). This study was approved by the Jen-Ai Medical Foundation Dali Jen-Ai Hospital: Human Body Research Ethics Committee, approval number 110-96. A flow diagram of this study is shown in Figure 1.

### 2.2. Research Materials

This section discusses the 8-week experimental method and content of the intervention, as shown in Table 1. The physical activity intervention in this study was walking, using a wrist-worn Garmin device (trade name: vivosmart HR smart bracelet) that uses vibrations during walking to count steps and electronic computing products through the 3D accelerator records in the chip. The device can measure items including steps, distance, calorie consumption, exercise intensity, and sleep time per night. The Garmin device can store 4 weeks of data [41,42,43]. The participants in this study were able to assess the results of daily and weekend steps, distance, calorie expenditure, and exercise intensity (the built-in device of the smart bracelet calculates the average heart rate (HR) during a day’s activities) as well as sleep time per night (using the built-in device of the smart bracelet), and through a daily memory setting could complete a daily record of their walks before removing the pedometer before bedtime. The participants were gathered once a week to integrate data from their Garmin devices.

### 2.3. Testing Method

According the above purpose, this study attempts to address the following two questions: 1. Did the walking intervention promote a healthy lifestyle in the older adults? 2. How did health behaviors of older adults predict the formation of a health-promoting lifestyle after the walking intervention? According to the research question, the following research hypotheses were proposed. Hypothesis one: The intervention of walking contributed to a healthy lifestyle in the older adults. Hypothesis two: Health behaviors after walking intervention could effectively predict health-promoting lifestyles in older adults. Therefore, the following detection content was designed.

#### 2.3.1. Garmin Built-in Data

Garmin’s vivosmart HR is a smart bracelet with a built-in pedometer sensor and barometric altimeter that can distinguish the difference in energy consumption between walking on flat ground or climbing steps and can automatically count the number of steps, distance, and calories, exercise intensity (referring to the degree of body tension and muscle contraction during exercise, with heart rate (HR) generally being used as an exercise intensity indicator [28]). According to Coswig et al., the moderate exercise intensity of the older adults (≥65 years old) is equivalent to between 55% and 75% of the maximum heart rate (MHR = 220 – age) (approximately 82 to 111 beats per minute) [44]. According to the recommendations of the Sleep Foundation, older adults (≥65 years old) need about 7–8 h of sleep per night [45]. The data were synchronized to the user’s mobile phone. After 12:00 every night, all of the records in the vivosmart were reset to zero and recalculated, but the previous records were automatically stored and retained. Vivosmart HR provides the Android and iOS dual-system support app “Garmin Connect”. You can connect vivosmart HR to mobile devices via Bluetooth to adjust the related settings and stream data. This most advanced vivosmart HR sports smart bracelet is easy to operate and has multiple functions, automatic calculation, strong stability, and good accuracy. The study cutoff points, according to the classification criteria of the number of walks per day by Tudor-Locke et al., were sedentary (<2500 steps/day), low activity (2500–5000 steps/day), somewhat active (5000–8000 steps/day), active (8000–11,000 steps/day), and highly active (>11,000 steps/day) [46].

#### 2.3.2. Health Behavior Inventory

This study revised the Health Behavior Inventory-20 (HBI-20), which was a measure of male health behaviors [23]. Originally, factor 4 of the HBI-20 scale was aimed at male genital health care. In order to meet the needs of both male and female older adults who were participants in this study, factor 4 (regular physical activity) and questions 15 to 17 were revised, and the inventory was renamed as the Health Behavior Inventory (HBI) of Older Adults. The first draft of the inventory was completed, and 100 senior students (average age 68.5 ± 1.43 years old) of the College of Performing Arts were sampled for a pre-test, and the research team assisted on-site to fill in the inventory. Twenty items remained after item analysis was performed on the results of the pre-test, which were then divided into five factors. The reliability analysis of each factor was as follows: healthy diet habit (five items, explained variance for 17.51%, Cronbach’s α = 0.81); proper use of health care resources (six items, explained variance for 12.20%, Cronbach’s α = 0.77); avoid strong emotions, tension, and stress (three items, explained variance for 11.44%, Cronbach’s α = 0.75); regular physical activity (three items, explained variance for 8.65%, Cronbach’s α = 0.72); and avoid tobacco, alcohol, and drug use (three items, explained variance for 6.24%, Cronbach’s α = 0.74). The scale has good reliability, with a Cronbach’s alpha between 0.72 and 0.81. the participants rated each item on a five-point Likert scale, with the following response range: 1 = strongly disagree, 2 = slightly disagree, 3 = slightly agree, 4 = agree, and 5 = strongly agree [47]. A higher total score indicates a higher frequency of implementing healthy behaviors.

#### 2.3.3. Health-Promoting Lifestyle Inventory

This study revised “The development of Chinese version health-promoting lifestyle scale (HPLI)” published by Taiwan scholar Chen et al. in 1997 [48]. The first draft of the inventory was completed, and 100 senior students (average age 68.5 ± 1.43 years old) of the College of Performing Arts were sampled for a pre-test (same sample as HBI pre-test), and the research team assisted on-site to fill in the inventory. The first draft of the scale was compiled, and after the expert content validity, 100 senior students from the Taiwan College of Performing Arts were randomly sampled as pre-test objects. After the pre-test was completed, the internal consistency of the project analysis was high, so no questions were deleted. The total cumulative explained variance of the scale reached 71.22%, which had construct validity. Reliability analysis was performed in order to understand the consistency and stability of the scale, and the Cronbach’s α value was used to analyze the internal consistency of the items with the same factors. The Cronbach’s α value of the reliability analysis results was between 0.78 and 0.84. Factor analysis extracted six factors, including health responsibility (five items), physical activity (six items), nutrition (seven items), spiritual growth (five items), social support (six items), and stress management (five items), resulting in a total of 34 items and named the “Health-Promoting Lifestyle Scale” for older adults. This questionnaire used a five-point Likert scale with the following response intervals: 1 = strongly disagree, 2 = slightly disagree, 3 = slightly agree, 4 = agree, and 5 = strongly agree [47]. The total average score of all participants for each question was between 3.5 and 5 points, and the higher the average score, the greater the tendency for all of the participants who answered the question to agree.

### 2.4. Control Variables

This study only focuses on healthy older adults over the age of 65, and at the same time, implements walking physical exercise. Some of the older adults did not have a sufficient cognitive level to cooperate and participate in the established meetings in the study [38]. This was a variable that must be controlled in this study. Garmin’s vivosmart HR device was operated by a researcher as a consultation point, and the participants gathered weekly to share the portion of data from the aggregated Garmin device that needed to be controlled for this study. In addition, 120 participants conducted healthy living and diet training before the experiment, and met every week (those who could not meet at the scheduled time would be contacted by phone) to verbally distribute health promotion materials (including medical care, healthy diet, and stress relief). In order to reduce the interference in the process of this experiment, this is the control variable of this study.

### 2.5. Statistical Analysis

First, Cohen’s d was calculated to determine the effect size of the *t*-test [49]. JASP statistical software was used to detect the Cohen’s d effect size [50]. In general, Cohen’s d values between 0.2 and 0.5 are small effects, between 0.5 and 0.8 are moderate effects, and values above 0.8 are large effects. Cohen’s d for age was 0.347 in the two groups of participants in this study, which was a small effect size and indicated the adequacy of the sample size. All participants underwent pre- and post-test questionnaires (HBI and HPLI). The obtained data were subjected to descriptive statistical analysis of the standard deviation (SD) and mean (M) of the pre- and post-tests [51] and t-tests for steps, time, calories burned, exercise intensity and sleep time per night for males and females, and the overall significance level was set at *p* < 0.05. The Pearson product-difference correlation was used to compare the relationship between the factors of the two scales, and finally, multiple regression analysis was used to understand the explanatory power of the health behavior factors for health-promoting lifestyles. The data of this study were statistically analyzed using SPSS 20.0 software (IBM^®^, Armonk, NY, USA).

## 3. Results

### 3.1. Participant Background Variable Analysis

The background variables of the 120 recruited participants in this study include gender, age (years), weight (kg), height (cm), BMI (kg/m^2^), previous occupation, and current residence status, as shown in Table 2. There was no significant differences in mean age between the males and females. Comparing the average weight or height of the males and females to reach a significant difference, but comparing the BMI of the males and females, did not show a significant difference; this phenomenon will be discussed later. According to the statistics of the previous occupations of the participants, the males had mainly been involved in the construction industry, followed by the transportation industry, whereas the females mainly worked in the medical and health care industry, followed by restaurants or hotels. The findings show that these older adults had a wide range of previous occupations. Since the recruited participants all live in the city, they are all retired from the workplace. The current living conditions showed that both males and females mostly live with their spouses, followed by living with their spouse and children. In addition, the difference between males and females living alone was the largest, with the proportion of males being higher than that of females.

The relationship between the previous occupation of the older adults and the pre-test of Health Behavior Inventory (HBI) is shown in Table 3. In terms of occupation, the “medical and health care industry” occupation of the male and female older adults had the highest correlation with various factors of health behavior, followed by “cultural and educational institutions” and “public utilities”. In terms of gender, the “regular physical activity factor of HBI” correlation between the male and female older adults in each occupation was the lowest. “Regular physical activity” had the lowest correlation among male older adults, followed by “avoid tobacco, alcohol and drug use”. The female older adults were least correlated with “regular physical activity”. The above shows that different occupations and different genders have different degrees of correlation with health behaviors or habits.

### 3.2. Analysis of Daily Walking, Walking Distance, and Calorie Consumption

The average stride length of the males was 64.16 cm, and the average stride length of the females was 60.38 cm. After 8 weeks of the experiment, the average number of steps, distance walked, calories consumed, exercise intensity, and sleep time per night were analyzed by t-test difference, as shown in Table 4. The study found that there were significant differences between the males and females in terms of the average daily steps, walking distance, and calories burned from Monday to Saturday, and the results of analyzing males’ walking behavior were better than those of the females. Conversely, there were no significant differences in steps, distance walked, and calories burned between the males and females on Sunday. On average, the males walked about 7800 to 8700 steps per day from Monday to Friday, which was considered positive. The females walked about 7200 to 7500 steps, which was a bit active. On Saturday and Sunday, the average number of steps per day for males and females was about 4500 to 5600 steps, which represents low activity. Based on the above, walking every day from Monday to Friday was driven by group activities and special personnel monitoring, and the target number of walks could be reached. On Saturdays and Sundays, the older adults might have family gatherings, home breaks, travel or leisure activities, etc., and take fewer steps and consume fewer calories. In addition, from Monday to Sunday, there was no significant difference in the exercise intensity and sleep time of the males and females. The walking intensity and *HR* of the male and female older adults from Monday to Friday reached moderate exercise intensity, at 102.61 to 107.61 beats/min, and the sleep time was between 7 and 8 h every night. On Saturday and Sunday, the amount of activity decreased, and the average *HR* of the day remained at 68.71 to 66.25 beats/min. Sleep time was still maintained at between 7 and 8 h every night.

### 3.3. Analysis of Participants’ Pre- and Post-Test HBI

The pre- and post-test HBIs for all participants in this study are shown in Table 5. It was found that after 8 weeks of the pedometer intervention for the males and females, most factors were significantly different from the items. Among them, there was no significant difference between factor 5 and items 18 to 20 for females, indicating that the pre- and post-test responses of female older adults to “avoid tobacco, alcohol, and drug use” were quite consistent, all of which were to avoid tobacco, alcohol, and drug use. The overall HBI scale showed that both males (*t* = 11.41, *p* < 0.05) and females (*t* = 14.36, *p* < 0.05) had the largest pre- and post-test differences in the “regular physical activity” factor. In addition, although males achieved significant differences pre- and post-test in the “avoid tobacco, alcohol, and drug use” factor, the post-test results showed that males had a significant improvement in this factor. In particular, there was no significant difference between the pre- and post-tests of the factors of “avoid tobacco, alcohol, and drug use” for the female older adults, but the average values of the pre- and post-tests showed that they all presented a very agreeable option, showing less use of tobacco, alcohol, and drugs.

### 3.4. Analysis of Participants’ Pre- and Post-Test HPLI

The HPLI pre- and post-test results of all participants in this study are shown in Table 6. It was found that after 8 weeks of the pedometer intervention for the males and females, most factors were significantly different from the items. Among them, the males and females did not have significant differences in the pre- and post-tests of questions 2 to 4, and the average of the pre- and post-tests showed that they strongly agreed, indicating that the older adults of both sexes have a high degree of recognition of health responsibility factors, will discuss personal health issues with health experts, seek the opinion of a second person when in doubt, and, at this age, they will also check their bodies at least once every year to identify any physical changes.

The male older adults had the largest pre- and post-test differences in the physical activity factor. From the pre- and post-test averages, the male participants all highly agreed that physical activity was the best way to promote health. After 8 weeks of the walking intervention, the males were able to perform 30 min or more of light to moderate physical activity at least three times a week. From the results of answering each question, the males unanimously agreed that exercise could reduce stress (*t* = 14.53, *p* < 0.05). The pre- and post-test differences in social support factors were the largest among the female older adults. The average pre- and post-test results showed that the female participants highly agreed that social support was the best way to promote health. After 8 weeks of the walking intervention, the females felt that they were closer with their friends. Exercise results in a good mood and one can receive the support of friends while exercising. From the results of answering each question, the females were unanimously in agreement that they would receive support from friends during exercise (*t* = 11.67, *p* < 0.05). The above results confirmed hypothesis one: the intervention of walking helps older adults to promote healthy lifestyles.

### 3.5. Pearson Product-Moment Correlation Analysis of the Post-Test Results of Various Factors between HBI and HPLI

Because each group was of an equal distance scale, this study used Pearson product-moment correlation analysis to assess the post-test results of each factor of the HBI and HPLI in the males and females.

In the post-test correlation analysis of each factor between the HBI and the HPLI in males, the correlation coefficient was between 0.60 and 0.92, which was significantly moderately and highly correlated, as shown in Table 7. After the 8-week walking intervention in males, there was a positive moderate–high correlation between healthy behaviors and a health-promoting lifestyle. Among them, the two-factor correlation coefficient between the “regular physical activity” of the HBI and “physical activity” of the HPLI was the highest at 0.92 (*p* < 0.05). Second, the correlation coefficient between the “regular physical activity” of the HBI and “stress management” of the HPLI was 0.89 (*p* < 0.05).

In the post-test correlation analysis of each factor between the HBI and the HPLI in females, the correlation coefficients were between 0.62 and 0.94, which were significantly moderately and highly correlated, as shown in Table 8. After the 8-week walking intervention in females, there was a positive moderate–high correlation between healthy behaviors and a health-promoting lifestyle. Among them, the highest correlation coefficient between the “regular physical activity” of the HBI and “social support” of the HPLI was 0.94 (*p* < 0.05). The second was the correlation coefficient between the “regular physical activity” of the HBI and “physical activity” of the HPLI, which was 0.91 (*p* < 0.05).

### 3.6. Explanatory Power of Various Factors of Health Behavior on Health-Promoting Lifestyle

After the Pearson product-moment correlation analysis, it was found that the factors had a high correlation between the health behaviors of the male and female older adults and the formation of a health-promoting lifestyle. To obtain the explanatory power, multiple regression analysis was used first to test the statistical significance of this regression model. The dependent and independent variables of the males and females were significantly different according to the F test results, where male F = 192.178 * (*p* < 0.01) and female F = 173.464 * (*p* < 0.01). Finally, the forced entry method (independent variables considered in the regression model) was used to analyze the explanatory power of the dependent variables for the health-promoting lifestyle. 

The explanatory power of males’ health behaviors on health-promoting lifestyles reached a significant level, and the explained variance was R^2^ = 70.9% (*p* < 0.01). The independent variable had the greatest effect on the estimated value of regular physical activity (β = 0.624, *p* < 0.01), followed by healthy diet habits (β = 0.531, *p* < 0.01). The five independent variables β all had positive values, indicating that the independent variables had a positive impact on the health-promoting lifestyle of male older adults, as shown in Table 9.

The explanatory power of females’ health behaviors on health-promoting lifestyles reached a significant level, and the explained variance was R^2^ = 74.1% (*p* < 0.01). The independent variable had the greatest effect on the estimated value of regular physical activity (β = 0.606, *p* < 0.01), followed by healthy diet habits (β = 0.578, *p* < 0.01). The five independent variables β were all positive, indicating that the independent variables had a positive impact on the health-promoting lifestyle of female older adults, as shown in Table 10. These results validated hypothesis two: healthy behaviors after the walking intervention could effectively predict health-promoting lifestyles in older adults.

## 4. Discussion

In this study, healthy older adults were sampled from a certain community in Taipei City, and estimates and inferences were made for all participants. Although it was not a comprehensive survey and experiment, the experimental results of 120 healthy older adults have been able to reflect the overall health information of healthy older adults in Taipei City. In the background variables of the healthy older adults in this study, there was a significant difference in the average weight and height between the males and females, but there was no significant difference in BMI between them. This phenomenon should focus on BMI, because BMI is more important than weight and height can better show health problems, but neither of them can diagnose the health status of the individual [52], so it will not affect the detection results of the smart bracelet. The main difference lies in the BMI level of the participants; some studies have pointed out that the BMI of older adults is negatively correlated with walking distance [53,54,55]. Although there were differences in pre-retirement occupations between the males and females, they had all worked in society, had general social cognition, and could use the smart bracelets that were popular in the market. Secondly, this study increased the social life of the older adults for group activities, which helped them to improve their quality of life, to find companionship and emotional support besdies the spouses or children they live with, and brought many health benefits to the older adults. These results were consistent with many previous studies [56,57,58,59].

After 8 weeks of the pedometer intervention, the difference between the participants was statistically significant (*p* < 0.05), which was consistent with the findings of many scholars [60,61,62,63]. After the pedometer intervention, the number of steps, walking distance, and calorie consumption in the male and female older adults on weekdays did improve their health-promoting lifestyles, which is consistent with many studies [64,65,66,67]. However, this study adopted a group walking method in large parks led by a specific person every Monday to Friday, and all participants performed their own activities on weekends. In fact, weekend exercise or physical activity were voluntary behaviors. Will weekends affect the results of the experiment? Haase confirmed that the effects of moderate-to-vigorous physical activity, whether spread out from Monday to Friday or concentrated on weekends, had no significant difference in health benefits [68]. Moreover, exercising from Monday to Friday can improve cognitive ability more than for those who only exercise on holidays [69]. This study arranged the days that the participants were to exercise each week, which invisibly helped the cognitive ability of the older adults, which was an additional effect of the study. The results of this study show that both male and female older adults can reach a heart rate of between 80 and 111 during walking exercise, which was in line with a moderate exercise intensity, which was consistent with the results of Coswig et al. [44]. At the same time, it also shows that from Monday to Sunday, the sleep time was maintained between 7 and 8 h every night, which was in line with the normal sleep time of older adults over 65 years old and was consistent with the research results of Suni and Singh [45].

In this study, walking among the older adults significantly improved health behaviors, and showed that frequent walking has a positive impact on the health of older adults [70]. According to a UK study, older adults can reduce their risk of developing dementia by 25% if they take 3800 steps/day, and by 50% if they walk 10,000 steps/day [71]. In addition, it has been found that older adults suffer from chronic diseases of hyperglycemia, and a fixed number of walks per day for many years can effectively control blood glucose [72]. Some people may question whether frequent walking among older adults will affect the joints, but studies have shown that walking in older adults has a positive lubricating and protective effect on bone joints [73,74,75]. Since walking has the effect of promoting health in older adults, how many steps should older adults walk every day? As early as 2011, it was reported that in older adults who were engaged in moderate-intensity walking, the number of steps taken by free-living activities per day was estimated to be about 7000–10,000 steps/day [46]. A goal of 10,000 steps/day has long been believed by the public to be necessary for health. In recent years, studies have shown that increases in the number of steps per day are associated with reduced mortality in older adults, confirming that the number of steps per day has increased, stabilizing at approximately 7500 steps/day, while the mortality rate has gradually decreased [76]. The World Health Organization (WHO) recommends that older adults engage in 2–2.5 hours of moderate exercise per week, which means walking 7000 or 8000 steps/day or 3 to 4 miles/day [77]. The results of the participants’ experiments in this study were consistent with the literature, confirming that the physical activity of walking has benefits in promoting health in older adults.

This study found that “regular physical activity” was the worst pre-test for HBI among male and female older adults before retirement. The “medical and health care industry” occupations of male and female older adults had a high degree of health behavior, which might itself be related to the occupation of medical health care and health behavior. This result has been confirmed [78,79,80], and in addition to the occupational nature of male older adults before retirement, smoking and drinking often become social tools, and this result was consistent with many studies [81,82,83]. The mean and *t*-test of the participants’ HBI showed that the ratio of tobacco, alcohol, and drug use in males was higher than in females [84,85,86,87], and healthy diet habits were lower in males than in the female older adults, which was consistent with the findings of other scholars [88,89,90]. Raptou and Papastefanou’s study pointed out that, in the short term, nicotine consumption increases energy expenditure and reduces appetite, and indeed the appetites of smokers were often lower than those of non-smokers and their healthy diet habits were also poor [91]. However, in this study, after 8 weeks of the walking intervention, the “healthy diet habits” and “avoid tobacco, alcohol, and drug use” factors in both male and female older adults were improved; that is, by avoiding tobacco, alcohol, and drug use and increasing the frequency of physical activity, the healthier the participants’ lifestyles became. This phenomenon is consistent with the research of many other scholars [92,93,94,95,96,97,98]. In terms of “regular physical activity”, the male and female older adults were able to exercise at least three days a week for more than 30 min each time and go outdoors or exercise in their spare time. In addition, in the post-test results of the HBI, it was found that the male and female older adults believed that they had a partner to exercise with, and obtaining companionship increased their exercise time, which was consistent with the research of many scholars [99,100,101].

The mean and t-test of the pre- and post-test responses of the participants’ HPLI showed that the male older adults believed that health-promoting lifestyles were most effective through physical activity and that physical activity was the most effective in reducing stress. These results indicate that male older adults could use physical activity or exercise to reduce stress and achieve a quality healthy lifestyle, which was consistent with many studies [102,103,104,105]. Female older adults believed that they could be praised by their friends when exercising. This social support was most effective in helping female older adults to form health-promoting lifestyles. In addition, female older adults had higher access to social support (family, spouses, children, or friends) and were able to engage in higher levels of physical activity, which means greater support, and were more likely to be active because of this support. Greater enjoyment of physical activity in female older adults also made females more motivated to engage in physical activity or exercise; this result was consistent with many prior studies [106,107,108,109,110]. In addition, scholars have also emphasized that regular walking can increase social opportunities and is also important for the health promotion of the elderly [111,112], because walking with friends or family will make exercise more interesting, provides the opportunity to make new friends in the process of exercise, encourages walking further, and when frequent meeting with friends takes place, older adults will not be out of touch with society and their health will be promoted [113]. Conversely, the main barriers to social support in older adults might be attributed to chronic loneliness or lack of interaction with spouses, friends, peers, and family members, resulting in insufficient social and communication skills and thus an inability to obtain information about health from the surrounding environment [114]. According to research, after the HBI and HPLI tests, older adults had a high degree of recognition for physical activity and social support, and social support was an important variable that affects the development of exercise habits in older adults. This view was consistent with many scholars. Therefore, the results of this study can induce most older adults to continue to exercise and develop positive habits [115,116,117,118,119].

According to the regression analysis results, the health behaviors of male older adults could effectively explain their health-promoting lifestyles, with 66.7% of males (R^2^ = 0.66.7) and 72.9% of females (R^2^ = 0.729) showing that improving healthy behaviors had the effect of forming health-promoting lifestyles. The regular physical activity factor of “healthy behavior” of male and female older adults was the most effective explanation for the formation of a “health-promoting lifestyle”, followed by “healthy diet habits” optimizing health to a high degree. Both factors were also highly explanatory for health-promoting lifestyles, which is consistent with many previous studies [120,121,122,123,124]. It was shown that the healthy behaviors of the older adults improved after the experiment and that these behaviors developed healthy diets and physical activity comprising a health-promoting lifestyle, which was consistent with the results of many studies [18,125,126,127,128]. From the above results, a healthy diet and physical activity are closely related to promoting health. Diet is primarily focused on nutritional advice, and in addition to adopting a healthy diet, physical activity (or exercise) is one of the most important things an individual can do to improve their health. Physical activity also plays a critical and complementary role in promoting health and preventing disease, including many chronic diet-related diseases. It is of great help to the promotion of the quality of life of older adults. Health promotion in older adults through healthy diet and increased physical activity is thus an important health issue at present.

This study has several limitations. The research sample was a convenience sample of the older adults recruited from only one community in Taipei City, so the inference of these results to older adults in other communities may be limited. The study is also limited in terms of its broad inferences to male and female healthy older adults, and future research should aim to devise an approach that encompasses a wider age range and design different types of physical activity to compare with each other. There were researchers on site responsible for roll call, recording, monitoring, etc., therefore errors caused by human factors were also one of the limitations of this study. The smart bracelet was the primary tool used in this research, and it is possible that the step counting of the smart bracelet was not accurate, or the smart bracelet could have certain limitations. Differences between the one-group pre- and post-test results could be caused by factors other than those assessed in the intervention, such as older adults feeling fatigued, losing interest, or becoming sick during a break in the experiment. In addition, this study focused primarily on a cross-sectional study of health-promoting lifestyles in older adults and was unable to provide definitive evidence of causality for the study variables.

## 5. Conclusions

In this study, specially assigned personnel instructed a sample of older adults to wear smart bracelets and monitored their walking activity for 8 weeks. HBI and HPLI questionnaires were used to conduct pre-tests and post-tests for all participants. The results found that the walking intervention helped the older adults improve blood circulation, strengthen muscles and bones, lubricate joints, improve mental health, and promote social interaction. All of the above benefits of walking positively influence the health behaviors of older adults for a health-promoting lifestyle. It is recommended that older adults make walking a part of their everyday routine; future study can use different types of physical activity to compare the effects and differences in older adults. In the future, research can be carried out on different age levels to determine the health behaviors and health-promoting lifestyles of adults of different age groups in a more subtle way. The value of this research is that it can help promote the health of older adults and provides public health advice.

## Figures and Tables

**Figure 1 ijerph-20-01628-f001:**
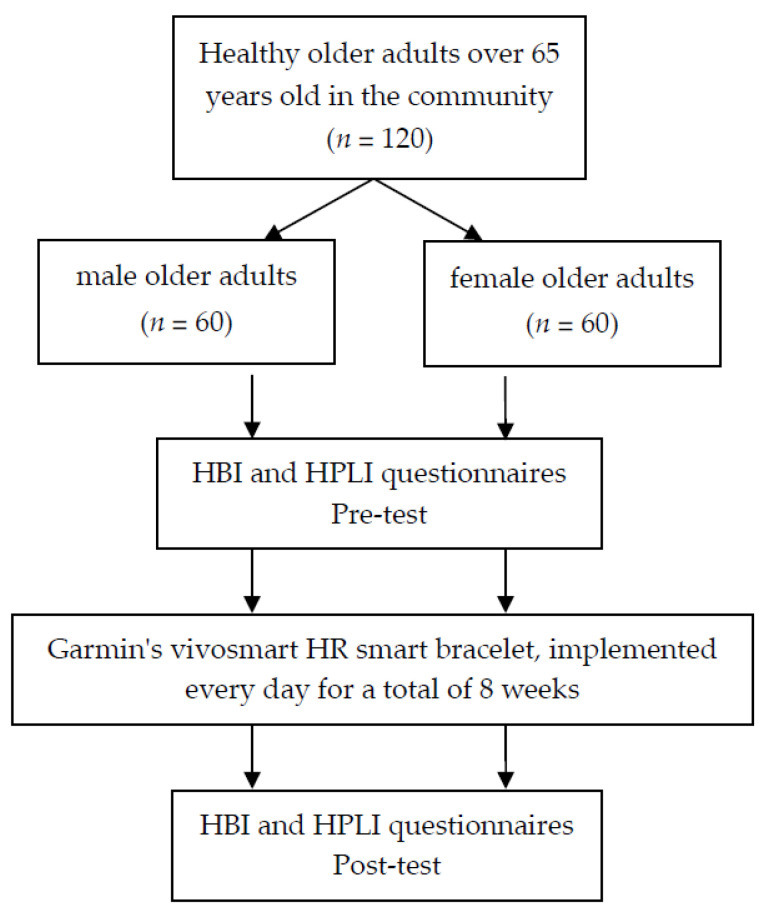
Participants and study flow diagram.

**Table 1 ijerph-20-01628-t001:** Experimental method and content.

Group	Method	Content
Male and female	Wristband walker	Dates: 8 weeks in total. Time: Monday to Friday, fixed 3:30–6:00 pm (during this period of time, participants could choose their own exercise time, and there was a roll call and record on site). Location: Walked together in the park from Monday to Friday, interviewer present for monitoring. Self-administered on Saturday and Sunday. Data collection: Participants recorded the total number of steps, time, calorie consumption, sleep time per night (using the built-in device of the smart bracelet) by themselves every day. They gathered in the park from 3:30 to 4:00 pm every other Monday for the test team to collect the data from all participants for the previous week.

Note: Wristband walker: Garmin vivosmart HR device (trade name: vivosmart HR smart bracelet)(Garmin, Taoyuan City, Guishan District, Taiwan (ROC)) [34].

**Table 2 ijerph-20-01628-t002:** Sociodemographic characteristics of the sample.

Variables	Male (n = 60)	Female (n = 60)	*t*-Value	*p*-Value
	M ± SD	M ± SD		
Age (years)	71.6 ± 1.25	72.3 ± 1.28	1.281	0.09
Weight (kg)	69.13 ± 4.82	62.56 ± 5.17	3.675 ^*^	0.00
Height (cm)	170.76 ± 7.18	159.94 ± 6.34	7.856 ^*^	0.00
BMI (kg/m^2^)	23.71 ± 1.65	24.46 ± 1.73	−1.95	0.07
**Previous occupation (retired)**	%	%	*t*-value	*p*-value
Cultural and educational institutions	8.5	9.5	−1.15	0.14
Public utilities	8.0	9.5	1.56	0.08
Restaurant or hotel business	8.0	11.0	−3.42 *	0.01
Transportation industry	13.0	5.5	5.82 *	0.00
Construction industry	17.0	8.5	6.17 *	0.00
Manufacturing	7.5	5.5	2.06	0.07
News advertising	4.5	8.0	−3.54 *	0.00
Medical and health care industry	5.5	13.5	−6.58 *	0.00
Entertainment industry	8.5	8.0	1.07	0.26
Service industry	10.5	10.5	0.84	0.54
General business	9.0	10.5	−1.635	0.09
**Current residence status**	%	%	*t*-value	*p*-value
Live alone	14.0	3.0	10.65	0.00
Live with spouse	44.0	49.0	−3.44 *	0.01
Living with spouse and children	39.5	47.0	−9.39 *	0.00
Live with friends	2.5	1.0	1.15	0.13

* *p* < 0.05. Means ± standard deviations presented as M ± SD.

**Table 3 ijerph-20-01628-t003:** Pearson product-difference correlation analysis of the factors of Health Behavior Inventory and occupational codes (males/females).

	A	B	C	D	E	F	G	H	I	J	K
**Factor 1**	0.38 */ 0.55 *	0.44 */ 0.52 *	0.33 */ 0.38 *	0.28 */ 0.31 *	0.28 */ 0.37 *	0.41 */ 0.52 *	0.48 */ 0.54 *	0.47 */ 0.66 *	0.38 */ 0.43 *	0.31 */ 0.35 *	0.33 */ 0.46 *
**Factor 2**	0.46 */ 0.59 *	0.53 */ 0.56 *	0.47 */ 0.43 *	0.34 */ 0.35 *	0.31 */ 0.42 *	0.56 */ 0.59 *	0.61 */ 0.67 *	0.65 */ 0.73 *	0.45 */ 0.51 *	0.37 */ 0.43 *	0.41 */ 0.52 *
**Factor 3**	0.45 */ 0.52 *	0.49 */ 0.51 *	0.48 */ 0.39 *	0.38 */ 0.43 *	0.45 */ 0.41 *	0.44 */ 0.53 *	0.47 */ 0.58 *	0.66 */ 0.69 *	0.32 */ 0.35 *	0.48 */ 0.51 *	0.40 */ 0.47 *
**Factor 4**	0.34 */ 0.33 *	0.41 */ 0.40 *	0.34 */ 0.31 *	0.29 */ 0.35 *	0.27 */ 0.34 *	0.37 */ 0.35 *	0.39 */ 0.34 *	0.41 */ 0.36 *	0.35 */ 0.32 *	0.30 */ 0.27 *	0.34 */ 0.31 *
**Factor 5**	0.45 */ 0.66 *	0.43 */ 0.49 *	0.37 */ 0.57 *	0.31 */ 0.45 *	0.29 */ 0.56 *	0.24 */ 0.53 *	0.43 */ 0.59 *	0.65 */ 0.78 *	0.31 */ 0.52 *	0.33 */ 0.47 *	0.27 */ 0.33 *

* *p* < 0.05. The following were the occupational codes of this study: A as cultural and educational institutions, B as public utilities, C as restaurant or hotel, D as transportation industry, E as construction industry, F as manufacturing, G as news advertising, H as medical and health care industry, I as entertainment industry, J as service industry, K as general business. The following were the factors of HBI: Factor 1: Healthy diet habit, Factor 2: Proper use of health care resources, Factor 3: Avoid strong emotions, tension and stress, Factor 4: Regular physical activity, Factor 5: Avoid tobacco, alcohol and drug use. The data shows males/females.

**Table 4 ijerph-20-01628-t004:** Descriptive statistics, *t*-test of average stride, steps, distance walked, and calorie intake for males and females.

Day	Males (n = 60)	Females (n = 60)	*t*-Value	*p*-Value
	M ± SD	M ± SD		
Stride, cm	64.16 ± 4.25	60.38 ± 4.41	3.62 *	0.01
Monday, steps	7951.71 ± 153,42	7218.49 ± 114.73	4.76 *	0.00
Distance, km	5.10 ± 0.27	4.36 ± 0.31	3.62 *	0.01
Calories, kcal	467.56 ± 17.32	422.73 ± 13.64	4.76 *	0.00
Walking intensity—HR, beats/min	104.45 ± 2.84	103.92 ± 2.57	1.35	0.09
Sleep, hours	7.23 ± 0.51	7.18 ± 0.46	1.24	0.11
Tuesday, steps	8613.55 ± 142.81	7261.58± 146.25	10.48 *	0.00
Distance, km	5.53 ± 0.37	4.38 ± 0.29	5.13 *	0.00
Calories, kcal	485.49 ± 17.62	431.44 ± 15,77	5.16 *	0.00
Walking intensity—HR, beats/min	105.17 ± 2.56	104.63 ± 2.24	1.48	0.09
Sleep, hours	7.57 ± 0.46	7.46 ± 0.31	1.17	0.23
Wednesday, steps	8535.15 ± 166.25	7434.82 ± 136.41	11.46 *	0.00
Distance, km	5.48 ± 0.37	4.49 ± 0.25	4.95 *	0.00
Calories, kcal	502.16 ± 22.72	443.85 ± 17.33	6.76 *	0.00
Walking intensity—HR, beats/min	107.61 ± 2.38	105.59 ± 2.44	1.43	0.09
Sleep, hours	7.81 ± 0.47	7.67 ± 0.39	1.39	0.10
Thursday, steps	8342.21 ± 119.15	7436.42 ± 121.28	9.17 *	0.00
Distance, km	5.35 ± 0.27	4.49 ± 0.31	4.02 *	0.00
Calories, kcal	492.36 ± 16.83	447.62 ± 15.53	4.73 *	0.00
Walking intensity—HR, beats/min	103.93 ± 2.85	102.61 ± 2.63	1.41	0.09
Sleep, hours	7.55 ± 0.41	7.51 ± 0.39	1.06	0.23
Friday, steps	7812.52 ± 96.44	7347.75 ± 101.17	5.33*	0.00
Distance, km	5.01 ± 0.18	4.44 ± 0.23	3.45 *	0.01
Calories, kcal	471.85 ± 11.24	442.59 ± 11.08	4.64 *	0.00
Walking intensity- HR, beats/min	104.92 ± 2.59	104.11 ± 2.33	1.15	0.21
Sleep, hours	7.86 ± 0.42	7.29 ± 0.38	1.76	0.08
Saturday, steps	5574.26 ± 185.29	5158.92 ± 172.31	3.25 *	0.02
Distance, km	3.58 ± 0.27	3.12 ± 0.19	3.21 *	0.04
Calories, kcal	332.93 ± 10.25	309.17 ± 9.43	3.23 *	0.03
Walking intensity—HR, beats/min	68.71 ± 5.91	67.37 ± 4.76	1.15	0.17
Sleep, hours	8.17 ± 0.76	7.85 ± 0.64	2.23	0.06
Sunday, steps	4652.22 ± 174.53	4583.65 ± 125.34	1.83	0.08
Distance, km	2.98 ± 0.19	2.77 ± 0.23	1.16	0.12
Calories, kcal	271.45 ± 12.78	255.37 ± 10.49	1.54	0.09
Walking intensity—HR, beats/min	67.14 ± 4.85	66.25 ± 4.51	1.13	0.18
Sleep, hours	7.94 ± 0.67	7.75 ± 0.59	1.86	0.08

* *p* < 0.05. Means ± standard deviations presented as M ± SD. Stride refers to the distance of each step when walking, also known as “step distance”, stride = distance/steps. HR refers to the heart rate. The moderate walking intensity of the older adults about 82 to 111 beats/minute. Sleep refers to the number hours of sleep per night.

**Table 5 ijerph-20-01628-t005:** Factors and items of Health Behavior Inventory.

Factors and Items	Males (n = 60)			Females (n = 60)		
	Pre-Test M ± SD	Post-Test M ± SD	*t*-Value (*p*-Value)	Pre-Test M ± SD	Post-Test M ± SD	*t*-Value (*p*-Value)
Factor 1: Healthy diet habit	3.09 ± 0.28	4.02 ± 0.19	−8.01 * (0.001)	3.86 ± 0.21	4.16 ± 0.13	−3.69 * (0.013)
1. I control the amount of fat I eat.	3.28 ± 0.26	4.16 ± 0.18	−7.69 * (0.002)	3.79 ± 0.26	4.14 ± 0.16	−3.75 * (0.011)
2. I control the amount of salt I eat.	3.13 ± 0.29	4.01 ± 0.21	−7.73 * (0.002)	3.71 ± 0.28	4.13 ± 0.17	−4.13 * (0.008)
3. I avoid eating large amounts of sugar.	3.15 ± 0.27	4.07 ± 0.16	−8.87 * (0.001)	3.85 ± 0.22	4.23 ± 0.11	−4.22 * (0.007)
4. I avoid chips and fried foods.	2.81 ± 0.33	3.76 ± 0.25	−8.95 * (0.001)	3.96 ± 0.20	4.16 ± 0.15	−3.18 * (0.015)
5. I control the amount of red meat I eat.	3.08 ± 0.34	4.09 ± 0.17	−9.09 * (0.001)	3.98 ± 0.19	4.12 ± 0.18	−1.12 (0.102)
Factor 2: Proper use of health care resources	3.11 ± 0.30	4.03 ± 0.19	−8.65 * (0.001)	3.11 ± 0.26	4.10 ± 0.16	−9.68 * (0.001)
6. I take the medicine according to the prescription.	3.31 ± 0.27	4.09 ± 0.17	−6.64 * (0.004)	3.34 ± 0.22	4.11 ± 0.17	−7.36 * (0.0011)
7. I go to all my scheduled health care appointments.	2.94 ± 0.31	3.87 ± 0.24	−8.97 * (0.001)	2.98 ± 0.31	3.98 ± 0.21	−9.61 * (0.0003)
8. I have dental exams every year.	2.93 ± 0.33	3.91 ± 0.24	−9.01 * (0.001)	2.95 ± 0.34	3.95 ± 0.22	−9.53 * (0.0003)
9. I take prescription medication only as directed by a health care provider.	3.19 ± 0.30	4.08 ± 0.19	−8.14 * (0.001)	3.18 ± 0.24	4.21 ± 0.12	−10.12 * (0.0002)
10. I take my blood pressure anytime.	3.13 ± 0.32	4.09 ± 0.18	−9.23 * (0.0005)	3.08 ± 0.29	4.13 ± 0.16	−10.24 * (0.0001)
11. I ask a healthcare provider when I have unfamiliar physical symptoms.	3.17 ± 0.31	4.15 ± 0.14	−9.31 * (0.0004)	3.15 ± 0.26	4.19 ± 0.14	−10.16 * (0.0002)
Factor 3: Avoid strong emotions, tension and stress	3.35 ± 0.22	3.87 ± 0.25	−4.34 * (0.005)	3.79 ± 0.25	4.00 ± 0.19	−3.15 * (0.037)
12. I get irritated and mad when waiting in lines.	3.31 ± 0.25	3.87 ± 0.21	−6.23 * (0.0017)	3.74 ± 0.27	3.98 ± 0.21	−3.19 * (0.018)
13. I get angry and annoyed when I am caught in traffic.	3.34 ± 0.23	3.85 ± 0.22	−5.24 * (0.004)	3.65 ± 0.29	3.92 ± 0.23	−3.21 * (0.012)
14. Things build up inside until I lose my temper.	3.39 ± 0.22	3.89 ± 0.20	−5.13 * (0.004)	3.97 ± 0.24	4.11 ± 0.15	−1.16 (0.17)
Factor 4: Regular physical activity	2.95 ± 0.26	4.12 ± 0.16	−11.41 * (0.0001)	2.75 ± 0.28	4.05 ± 0.18	−14.36 * (0.0001)
15. I exercise at least three days a week for more than 30 min each time.	2.56 ± 0.34	4.05 ± 0.23	−18.42 * (0.0001)	2.34 ± 0.37	4.01 ± 0.17	−19.26 * (0.0001)
16. I often have partner who exercises with me.	3.11 ± 0.25	4.12 ± 0.18	−10.17 * (0.0001)	2.89 ± 0.26	4.08 ± 0.15	−14.08 * (0.0001)
17. When I am free, I think about being outdoors or exercise.	3.17 ± 0.21	4.18 ± 0.17	−10.23 * (0.0001)	3.02 ± 0.24	4.05 ± 0.16	−10.17 * (0.0001)
Factor 5: Avoid tobacco, alcohol and drug use	3.14 ± 0.24	3.95 ± 0.17	−7.81 * (0.0009)	4.12 ± 0.13	4.22 ± 0.11	−0.85 (0.34)
18. I don’t smoke.	3.01 ± 0.27	3.89 ± 0.19	−8.58 * (0.0005)	4.31 ± 0.11	4.44 ± 0.09	−0.96 (0.21)
19. I do not use recreational drugs.	3.54 ± 0.21	4.11 ± 0.15	4.92 * (0.007)	4.03 ± 0.17	4.12 ± 0.16	−0.82 (0.33)
20. I don’t use alcoholic beverages.	2.87 ± 0.25	3.84 ± 0.20	−9.81 * (0.0001)	4.01 ± 0.19	4.09 ± 0.18	−0.77 (0.39)

* *p* < 0.05. Pre- and post-tests values presented as means ± standard deviations (M ± SD). *t* test value presented as t-value (*p*-value).

**Table 6 ijerph-20-01628-t006:** Factors and items of the HPLI.

Factors and Items		Males (n = 60)			Females (n = 60)		
		Pre-Test M ± SD	Post-Test M ± SD	*t*-Value (*p*-Value)	Post-Test M ± SD	Pre-Test M ± SD	*t*-Value (*p*-Value)
Factor 1: Health responsibility		3.87 ± 0.27	4.09 ± 0.15	−3.76 * (0.015)	3.92 ± 0.25	4.16 ± 0.11	−3.31 * (0.037)
1	Watch TV programs about improving health.	3.99 ± 0.21	4.19 ± 0.10	−3.84 * (0.013)	3.98 ± 0.21	4.28 ± 0.08	−3.97 * (0.007)
2	Discuss my health concerns with a health professional.	4.00 ± 0.19	4.11 ± 0.12	−1.07 (0.124)	4.06 ± 0.16	4.17 ± 0.11	−1.22 (0.094)
3	Get a second opinion when I suspect advice given by my healthcare professional.	3.97 ± 0.22	4.08 ± 0.16	−1.18 (0.116)	4.01 ± 0.18	4.15 ± 0.13	−1.43 (0.075)
4	Examine my body at least once a year to detect any physical changes.	3.91 ± 0.25	4.05 ± 0.17	−1.55 (0.085)	3.97 ± 0.23	4.08 ± 0.17	−1.31 (0.087)
5	Attend educational programs on health care.	3.47 ± 0.35	4.02 ± 0.19	−7.16 * (0.0003)	3.59 ± 0.32	4.11 ± 0.15	−6.54 * (0.0005)
Factor 2: Physical activity		3.23 ± 0.27	4.24 ± 0.10	−11.15 * (0.0001)	3.19 ± 0.22	4.09 ± 0.12	−9.82 * (0.0002)
6	Exercise vigorously for 30 min or more at least three times a week.	3.24 ± 0.27	4.21 ± 0.13	−9.68 * (0.0002)	3.19 ± 0.24	4.18 ± 0.08	−9.25 * (0.0003)
7	Participate in physical activity at a mild to moderate level.	3.20 ± 0.28	4.24 ± 0.11	−11.71 * (0.0001)	3.15 ± 0.26	4.07 ± 0.13	−8.41 * (0.0007)
8	Do stretching exercises at least 3 times a week.	3.19 ± 0.29	4.25 ± 0.16	−11.58 * (0.0001)	3.21 ± 0.23	4.05 ± 0.15	−8.23 * (0.0009)
9	Participate in leisure physical activities.	3.24 ± 0.27	4.19 ± 0.14	−10.26 * (0.0001)	3.15 ± 0.26	4.04 ± 0.17	−8.04 * (0.0012)
10	Exercise while doing daily activities.	3.18 ± 0.29	4.25 ± 0.10	−11.79 * (0.0001)	3.17 ± 0.25	4.13 ± 0.11	−9.61 * (0.0002)
11	Checking my pulse rate while exercising.	3.30 ± 0.23	4.31 ± 0.07	−10.23 * (0.0001)	3.25 ± 0.22	4.06 ± 0.13	−7.57 * (0.0016)
Factor 3: Nutrition		3.65 ± 0.23	4.08 ± 0.12	−4.77 * (0.0046)	3.81 ± 0.21	4.21 ± 0.09	−4.30 * (0.0081)
12	Choose a diet low in fat, saturated fat and cholesterol.	3.59 ± 0.29	4.13 ± 0.11	−5.66 * (0.0011)	3.79 ± 0.25	4.25 ± 0.08	−4.61 * (0.0062)
13	Limit sugar intake and foods that contain sugar (sweet).	3.81 ± 0.21	4.11 ± 0.11	−4.23 * (0.0093)	3.85 ± 0.21	4.27 ± 0.07	−4.35 * (0.0075)
14	Eat 1.5 to 4 bowls of staple food, including cereals, rice and noodles daily.	3.66 ± 0.23	4.08 ± 0.13	−4.41 * (0.0065)	3.78 ± 0.27	4.19 ± 0.13	−3.73 * (0.028)
15	Eat 2 to 4 fist-sized pieces of fruit daily.	3.61 ± 0.20	3.95 ± 0.18	−3.54 * (0.032)	3.81 ± 0.23	4.11 ± 0.17	−3.45 * (0.037)
16	Eat 3 to 5 plates of vegetables daily.	3.69 ± 0.23	4.05 ± 0.14	−4.16 * (0.011)	3.82 ± 0.22	4.15 ± 0.15	−3.92 * (0.022)
17	Drink 350 to 500 mL of dairy daily.	3.53 ± 0.31	4.14 ± 0.10	−5.33 * (0.0028)	3.79 ± 0.26	4.20 ± 0.10	−4.12 * (0.015)
18	Eat 115–300 g based on meat, chicken, fish, dried beans, eggs and bean groups daily.	3.68 ± 0.22	4.09 ± 0.12	−4.37 * (0.0074)	3.86 ± 0.19	4.26 ± 0.07	−4.42 * (0.0063)
Factor 4: Spiritual growth		3.58 ± 0.21	4.18 ± 0.10	−5.45 * (0.0026)	3.63 ± 0.23	4.19 ± 0.11	−5.25 * (0.0054)
19	Get enough sleep.	3.63 ± 0.19	4.16 ± 0.12	−4.87 * (0.0062)	3.66 ± 0.22	4.14 ± 0.13	−4.54 * (0.0076)
20	Feeling I am growing and changing in a positive way.	3.57 ± 0.21	4.11 ± 0.16	−4.65 * (0.0074)	3.60 ± 0.25	4.15 ± 0.12	−5.33 * (0.0047)
21	Believe that my life has a purpose.	3.61 ± 0.20	4.18 ± 0.12	−5.39 * (0.0031)	3.78 ± 0.18	4.26 ± 0.06	−4.81 * (0.0064)
22	Looking to the future.	3.64 ± 0.19	4.29 ± 0.09	−9.42 * (0.0004)	3.69 ± 0.21	4.23 ± 0.07	−5.26 * (0.0053)
23	Concentrate on pleasant thoughts before bed.	3.41 ± 0.25	4.15 ± 0.13	−8.38 * (0.0006)	3.42 ± 0.28	4.16 ± 0.12	−7.49 * (0.0012)
Factor 5: Social support		3.45 ± 0.22	4.26 ± 0.09	−8.75 * (0.0005)	3.32 ± 0.22	4.21 ± 0.13	−9.28 * (0.00034)
24	Discuss exercise patterns with friends.	3.31 ± 0.26	4.17 ± 0.12	−9.47 * (0.0004)	3.26 ± 0.26	4.12 ± 0.16	−8.51 * (0.00057)
25	I would compliment a friend’s athleticism.	3.33 ± 0.25	4.16 ± 0.14	−9.59 * (0.0003)	3.21 ± 0.28	4.15 ± 0.14	−10.97 * (0.0001)
26	Exercise with friends.	3.35 ± 0.25	4.27 ± 0.09	−10.96 * (0.0001)	3.23 ± 0.27	4.17 ± 0.13	−11.09 * (0.0001)
27	I will show concern to my friends.	3.62 ± 0.18	4.24 ± 0.10	−7.34 * (0.0009)	3.54 ± 0.19	4.25 ± 0.11	−7.86 * (0.0008)
28	Playing sports with friends makes me happy.	3.49 ± 0.22	4.36 ± 0.07	−9.81 * (0.0002)	3.37 ± 0.23	4.32 ± 0.07	−11.53 * (0.0001)
29	Get support from friends while exercising.	3.58 ± 0.19	4.37 ± 0.06	−7.12 * (0.0011)	3.33 ± 0.24	4.24 ± 0.11	−11.67 * (0.0001)
Factor 6: Stress management		3.45 ± 0.23	4.28 ± 0.10	−8.46 * (0.0005)	3.47 ± 0.25	4.21 ± 0.11	−7.16 * (0.0018)
30	Accepting things I can’t change in my life.	3.39 ± 0.26	4.19 ± 0.13	−8.38 * (0.0007)	3.35 ± 0.30	4.11 ± 0.15	−7.22 * (0.0014)
31	Feel satisfied and calm with yourself.	3.62 ± 0.21	4.24 ± 0.11	−7.59 * (0.0011)	3.59 ± 0.23	4.14 ± 0.14	−4.79 * (0.0093)
32	Exercise makes me less stressed.	3.34 ± 0.28	4.53 ± 0.07	−14.53 * (0.0001)	3.44 ± 0.26	4.32 ± 0.07	−9.94 * (0.0003)
33	Finding that every day is exciting and challenging.	3.43 ± 0.25	4.33 ± 0.09	−10.26 * (0.0003)	3.41 ± 0.27	4.19 ± 0.12	−7.85 * (0.0008)
34	Calm myself to avoid fatigue.	3.49 ± 0.24	4.12 ± 0.15	−6.77 * (0.0015)	3.54 ± 0.24	4.31 ± 0.08	−7.39 * (0.0011)

* *p* < 0.05. Pre- and post-tests values presented as means ± standard deviations (M ± SD). *t* test value presented as t-value (*p*-value).

**Table 7 ijerph-20-01628-t007:** Correlation analysis of post-test results of various factors between the HBI and HPLI in males.

	HBI					
		Factor 1	Factor 2	Factor 3	Factor 4	Factor 5
		Healthy Diet Habit	Proper Use of Health Care Resources	Avoid Strong Emotions, Tension, Stress	Regular Physical Activity	Avoid Tobacco, Alcohol, and Drug Use
**HPLI**	**Factor 1: Health responsibility**	0.74 *	0.78 *	0.73 *	0.81 *	0.78 *
	**Factor 2: Physical activity**	0.80 *	0.83 *	0.85 *	0.92 *	0.75 *
	**Factor 3: Nutrition**	0.79 *	0.77 *	0.68 *	0.83 *	0.74 *
	**Factor 4: Spiritual growth**	0.68 *	0.73 *	0.81 *	0.84 *	0.66 *
	**Factor 5: Social support**	0.60 *	0.75 *	0.76 *	0.81 *	0.63 *
	**Factor 6: Stress management**	0.67 *	0.65 *	0.84 *	0.89 *	0.79 *

* *p* < 0.05.

**Table 8 ijerph-20-01628-t008:** Correlation analysis of post-test results of various factors between the HBI and HPLI in females.

	HBI					
		Factor 1	Factor 2	Factor 3	Factor 4	Factor 5
		Healthy Diet Habit	Proper Use of Health Care Resources	Avoid Strong Emotions, Tension, Stress	Regular Physical Activity	Avoid Tobacco, Alcohol, and Drug Use
**HPLI**	**Factor 1: Health responsibility**	0.86 *	0.87 *	0.84 *	0.86 *	0.87 *
	**Factor 2: Physical activity**	0.84 *	0.78 *	0.84 *	0.91 *	0.88 *
	**Factor 3: Nutrition**	0.87 *	0.81 *	0.75 *	0.84 *	0.78 *
	**Factor 4: Spiritual growth**	0.78 *	0.79 *	0.83 *	0.80 *	0.78 *
	**Factor 5: Social support**	0.83 *	0.83 *	0.86 *	0.94 *	0.62 *
	**Factor 6: Stress management**	0.79 *	0.84 *	0.84 *	0.87 *	0.74 *

* *p* < 0.05.

**Table 9 ijerph-20-01628-t009:** Multiple linear regression analysis of males.

Variables	β	R	R^2^	Adjusted R^2^	*p*
Healthy diet habits	0.531 *	0.842	0.709	0.707	<0.001
Proper use of health care resources	0.517 *				<0.002
Avoid strong emotions, tension, and stress	0.439 *				<0.006
Regular physical activity	0.624 *				<0.001
Avoid tobacco, alcohol, and drug use	0.481 *				<0.004

Independent variables: healthy diet habits; proper use of health care resources; avoid strong emotions, tension, and stress; regular physical activity; and avoid tobacco, alcohol, and drug use. Dependents: HPLI. * *p* < 0.01.

**Table 10 ijerph-20-01628-t010:** Multiple linear regression analysis of females.

Variables	β	R	R^2^	Adjusted R^2^	*p*
Healthy diet habits	0.578 *	0.861	0.741	0.739	<0.001
Proper use of health care resources	0.494 *				<0.003
Avoid strong emotions, tension, and stress	0.473 *				<0.004
Regular physical activity	0.606 *				<0.001
Avoid tobacco, alcohol, and drug use	0.455 *				<0.006

Independent variables: healthy diet habits; proper use of health care resources; avoid strong emotions, tension, and stress; regular physical activity; and avoid tobacco, alcohol, and drug use. Dependents: health-promoting lifestyle. * *p* < 0.01.

## Data Availability

The experimental results obtained real data about the study participants before and after training. The participants agreed with the data structure via confirmation, which can be disclosed upon reasonable request. All datasets on which the conclusions of the paper rely are available to editors, reviewers, and readers.

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
