# Peer review of "Promoting Healthy Behaviors in Older Adults to Optimize Health-Promoting Lifestyle: An Intervention Study"

_ijerph, 2023, doi:10.3390/ijerph20021628_

Round 1

Reviewer 1 Report

Dear Authors,
the manuscript is well structured. Statistical analysis is well done.

A suggestion for the next scientific works and epidemiological investigations: for the next work you could plan to include diffrent types of physical activity. This could be an interesting comparison.

Author Response

We are very grateful for the suggestions given by the reviewers, and our group of authors will continue to work hard to launch more types of research.

Reviewer 2 Report

Dear Editor,

Thank you for the opportunity to review the article entitled “Promoting Healthy Behaviors in Older Adults to Optimize 2 Health-Promoting Lifestyle: An Intervention Study”. In a study to improve the health behavior of the  older adults through a physical activity intervention and enhance a health-promoting lifestyle.

The following reviews are suggested:

- In the introduction identify similar studies and what this study brings that is different,

- Clarify in the title, abstract and methodology the type of study. Randomized controlled trial ? Quasi-experimental?

- In the method, mention who carried out the application of the evaluation instruments? The authors? If so, put it as a limitation of the study.

Congratulations for the excellent work.

Author Response

We are very grateful for the hard work of the reviewers, we benefited from the reviewers' valuable comments and revised the manuscript according to the reviewers' comments.

 In the introduction identify similar studies and what this study brings that is different,

Response: Revised complete, line 87-88, line 91, line 93-96, line 98-99.

- Clarify in the title, abstract and methodology the type of study. Randomized controlled trial ? Quasi-experimental?

Response: Revised complete, line 17, line 110-113.

- In the method, mention who carried out the application of the evaluation instruments? The authors? If so, put it as a limitation of the study.

Response: Revised complete, line 613-616.

Reviewer 3 Report

Congratulations to the authors of this article. It is a very interesting article.
This is an interesting article on a protocol for the Promotion of Healthy Behaviors in Older Adults. I have some doubts about the paper, which I will explain below:

Methodology:

Did you calculate the intensity performed by each of the subjects during walking?

Having significant differences in the basal characteristics of the subjects in weight and height, it would have been interesting to mark the intensity of the exercise in a better way, since having the smart bracelet is an easy thing to do.

Did you keep track of the hours of sleep? The subjects by exercising and eating healthier, surely that affected the quality of sleep, a fact that Garmin usually gives you.

Results:

In the results you repeat data in the table and in the text. It is interesting that in scientific articles you do not repeat data in figures/tables and in the text because it becomes too long and heavy to read.

If as you indicate there are differences in sociodemographic characteristics in weight and height, don't you think it can influence what you are measuring?

I would not put the p value for differences between the above occupations, I would try to perform correlations to find out if those professions influence the healthy habits of these people.

Conclusions:

In the conclusions you would have to be stricter, for example, not extrapolate to the whole world population, but focus on the population in which the intervention has been performed, in this case, in the city of Tapei.

Discussion:

There is a need for further discussion on the topic, as the article discusses many variables on health promotion such as diet, exercise, steps....

Other things:

It would be necessary to standardize the use of . or , for numerical data, in the text they appear with , and in the tables with .

Was there a follow-up after the investigation to know if these habits lasted over time, or did they simply remain in the weeks of the intervention?

It is obvious to me some significant differences, because if you do an intervention with exercise and diet, it is normal that during the intervention these factors "improve", the interesting thing is to see if this intervention achieves adherence after the intervention.

It is always good to show some image of intervention or graph in the article, there are many tables with variables that contribute little to the article and the data is repeated during the text.

Author Response

Thanks to the reviewers for their hard work, we sincerely appreciate the reviewers' valuable comments, and we revised the manuscript based on the reviewers' comments.

Did you calculate the intensity performed by each of the subjects during walking?

Response: Revised complete, line 91-92, line 167-170, line 191-195, line 338-345, line 352, line 481-486.

Having significant differences in the basal characteristics of the subjects in weight and height, it would have been interesting to mark the intensity of the exercise in a better way, since having the smart bracelet is an easy thing to do.

Response: Revised complete, line 191-197, line 284-286, line 481-488.

Did you keep track of the hours of sleep? The subjects by exercising and eating healthier, surely that affected the quality of sleep, a fact that Garmin usually gives you.

Response: Revised complete, line 195-197, line 338-345, line 352, line 513-516.

Results:

In the results you repeat data in the table and in the text. It is interesting that in scientific articles you do not repeat data in figures/tables and in the text because it becomes too long and heavy to read.

Response: Revised complete, line 280-290, line 295-296.

If as you indicate there are differences in sociodemographic characteristics in weight and height, don't you think it can influence what you are measuring?

Response: Revised complete, line line 284-286, line 481-488.

I would not put the p value for differences between the above occupations, I would try to perform correlations to find out if those professions influence the healthy habits of these people.

Response: Revised complete, line 297-306, line 312-319, line 540-546.

Conclusions:

In the conclusions you would have to be stricter, for example, not extrapolate to the whole world population, but focus on the population in which the intervention has been performed, in this case, in the city of Tapei.

Response: Revised complete, Delete line 627-629.

Discussion:

There is a need for further discussion on the topic, as the article discusses many variables on health promotion such as diet, exercise, steps....

Response: Revised complete, Add, line 481-488, line 510-516, line 540-546, line 584-589.

Other things:

It would be necessary to standardize the use of . or , for numerical data, in the text they appear with , and in the tables with .

Response: Revised complete, line 308-309, line 312-314, line 352-359.

Was there a follow-up after the investigation to know if these habits lasted over time, or did they simply remain in the weeks of the intervention?

Response: Revised complete, line 584-589.

It is obvious to me some significant differences, because if you do an intervention with exercise and diet, it is normal that during the intervention these factors "improve", the interesting thing is to see if this intervention achieves adherence after the intervention.

Response: Revised complete, line 584-589.

It is always good to show some image of intervention or graph in the article, there are many tables with variables that contribute little to the article and the data is repeated during the text.

Response: Revised complete, line 280-290, line 295-296.

Reviewer 4 Report

Many thanks to the journal for the invitation to review. Congratulations to the authors for the work done, attached are some comments to be considered. 

Author Response

Thanks to the reviewers for their hard work, we sincerely appreciate the reviewers' valuable comments, and we revised the manuscript based on the reviewers' comments.

TITLE AND ABSTRACT

- The title or abstract should give information on what kind of study it is about. It is also advisable to add the population where the study has been carried out.

Response: Revised complete, line 115-119, line 133-136.

- In addition, it would be relevant to justify why the intervention is targeted at this type of population and not others. Regardless of age, adherence to exercise depends on what motivates people to start or continue physical activity.

Response: Revised complete, line 93-96.

INTRODUCTION

- Very good start to the introduction, a good description of different area are covered, but is advisable to add the specified hypothesis of the study at the end of the introduction.

Response: Revised complete, line 103-107.

- It is recommended to emphasise the prevalence of obesity and associated morbidity and mortality.

o World Health Organization. U.S. Department of Health and Human Services. National Institute on Aging. National Institutes of Health;Global Health and Aging. 2011 :1–32. Available online: https://www.nia.nih.gov/sites/default/files/2017- 06/global_health_aging.pdf.

Response: Revised complete, line 49-51.

- It would be useful to talk about chronic diseases and older adults in the introduction. It would be interesting for the reader to get more information about this to put it into context, and if it is aimed at older adults, to mention the age range of this population and the benefits and drawbacks of lack of physical activity more broadly. On the other hand, to comment in more depth on the risks, benefits and disadvantages of physical exercise in this type of population.

Response: Revised complete, line 49-55, line 87-88, line 91-96, line 98-99.

- The benefits of using smart bracelets are discussed, but it would also be important

to add research by other authors based on physical activity. This allows us to assess

which variables should be taken into account when conducting the study.

Response: Revised complete, line 95-99.

MATERIAL AND METHODS

  1. Design

- It does not describe the design. Specify the type of study. For example, add this as

a subsection: “This quasi-experimental intervention study consisted of 120

participants, received dietary and activity recommendations, and were provided with

a smart device (vivosmart HR smart bracelet). Subjects were randomised following

the recommendations published by X [reference]”.

Response: Revised complete, line 115-118.

2.3. Testing method

- It is recommended to add the hypothesis at the end of the introduction.

Response: Revised complete, line 103-107.

  1. Participants

- All persons who have participated should be reported in the study, not only the

participants.

Response: Revised complete, line 121-123.

- How was the sample selected? through an advertisement, a reference site, a phone

call? Include eligibility and exclusion criteria and sources.

Response: Revised complete, line 115-117.

- How long did the study last? Indicate duration in months.

Response: Revised complete, line 126.

2.3.2. Health Behavior Inventory

- Include the methodology of the statistical analysis in this sub-section, applicable to the rest of the document.

Response: Revised complete, line 270-272.

Statistical Analysis

- Comments from previous contributions have been taken into account

Response: Revised complete, line 270-272.

RESULTS

- Table 2. include in brackets the percentage, so as not to repeat it repeatedly.

Response: Revised complete, line 308-310, line 312-314, line 352-359.

- Table 2. It is important that the tables are homogeneous, and that the information displayed always follows the same pattern. If 2 decimal places are included, it is important that this is always the case.

Response: Revised complete, line 308-319, line 352-359.

- Table 3. why is the p value crossed out.

Response: Revised complete, line 309, line 314, line 354, line 382, line 411, line 466, line 475

- Table 3. Include the unit of measurement in brackets, as in table 2. Review the rest of the document.

- The sample of men and women respectively is 30 or 60? check tables.

Response: Revised complete, line 308-310.

DISCUSSION

- Line 509: Do not use the first person plural. Applicable to the rest of the document.

Response: Revised complete, line 601.

CONCLUSIONES

- The conclusion includes aims and areas for improvement for future research.

Response: Revised complete, line 627-629, line 634-635.

REFERENCES

- References follow the indicated style.

Response: Revised complete, line 671-672, line 677-680, line 714-716, line 729-734, line 746, line 761-765, line 779-780, line 786-788, line 811-812, line 843-852.

Recommendations for Authors (will be shown to authors) The following questions do not substitute for specific comments made for authors. Please give further details in the comments for authors box below.

Does the introduction provide sufficient background and include all relevant references?

Must be improved, the introduction is very short and a the hypothesis of the study should be added at the end of the introduction.

Response: Revised complete, line 87-88, line 91-96, line 98-99, line 103-107.

Are all the cited references relevant to the research?

Yes

Is the research design appropriate?

Yes, but they still do not indicate the type of study in the title of the article.

Response: Revised complete, line 1.